# The miRNA-21-5p Payload in Exosomes from M2 Macrophages Drives Tumor Cell Aggression via PTEN/Akt Signaling in Renal Cell Carcinoma

**DOI:** 10.3390/ijms23063005

**Published:** 2022-03-10

**Authors:** Zhicheng Zhang, Junhui Hu, Moe Ishihara, Allison C. Sharrow, Kailey Flora, Yao He, Lily Wu

**Affiliations:** 1Department of Molecular and Medical Pharmacology, David Geffen School of Medicine, UCLA, Los Angeles, CA 90095, USA; zzc168439zzc@gmail.com (Z.Z.); junhuihu@mednet.ucla.edu (J.H.); mishihara@mednet.ucla.edu (M.I.); asharrow@mednet.ucla.edu (A.C.S.); kaileyflora@gmail.com (K.F.); 2Department of Microbiology, Immunology and Molecular Genetics, UCLA, Los Angeles, CA 90095, USA; yaohe@ucla.edu; 3Department of Urology, David Geffen School of Medicine, UCLA, Los Angeles, CA 90095, USA; 4Jonsson Comprehensive Cancer Center, David Geffen School of Medicine, UCLA, Los Angeles, CA 90095, USA

**Keywords:** miRNA-21-5p, exosomes, M2 macrophages, renal cell carcinoma, invasion, migration, PTEN/Akt

## Abstract

M2 macrophages in the tumor microenvironment are important drivers of cancer metastasis. Exosomes play a critical role in the crosstalk between different cells by delivering microRNAs or other cargos. Whether exosomes derived from pro-tumorigenic M2 macrophages (M2-Exos) could modulate the metastatic behavior of renal cell carcinoma (RCC) is unclear. This study found that M2-Exos promotes migration and invasion in RCC cells. Inhibiting miR-21-5p in M2-Exos significantly reversed their pro-metastatic effects on RCC cells in vitro and in the avian embryo chorioallantoic membrane in vivo tumor model. We further found that the pro-metastatic mechanism of miR-21-5p in M2-Exos is by targeting PTEN-3′UTR to regulate PTEN/Akt signaling. Taken together, our results demonstrate that M2-Exos carries miR-21-5p promote metastatic features of RCC cells through PTEN/Akt signaling. Reversing this could serve as a novel approach to control RCC metastasis.

## 1. Introduction

Renal cell carcinoma (RCC) ranks amongst the top ten most common cancer types, with over 400,000 newly diagnosed cases worldwide in 2018 [1]. Clear cell renal cell carcinoma (ccRCC) comprises almost 80% of RCC subtypes [2]. Approximately 30% of RCC patients present with metastatic disease, and 10–20% of patients with localized tumors would also develop metastasis later in the course of disease [3]. Unfortunately, metastatic RCC is resistant to current conventional therapies [4]. A better understanding of the mechanism of metastasis in RCC could be helpful to devise new therapeutic interventions.

Dictated by their front-line innate immune responses, macrophages are highly adaptable cells influenced heavily by environmental stimuli. Thus, macrophages are further categorized into the M1 and M2 subtypes to distinguish their functional activity [5]. The classical M1 macrophages display pro-inflammatory activities for clearing foreign pathogens and cancer cells. In contrast, tumor-associated macrophages (TAMs) are designated as M2 that provide numerous pro-tumorigenic functions [6,7], such as fostering an immunosuppressive environment and promoting cancer invasion and metastasis [8,9]. TAMs are found in all solid tumors, including RCC, and associated with RCC progression [10]. However, the specific mechanism by which M2 macrophages promote RCC metastasis is unknown.

Exosomes, small bilayer membrane vesicles with a diameter of 30 to 150 nm, play an essential role in intercellular crosstalk by delivering the instructional payload, such as mRNAs, microRNAs (miRNAs), long noncoding RNAs, and proteins [11,12]. For example, exosomes derived from M2 macrophages (M2-Exos) enhanced the migration and invasion capacities of colorectal cancer cells by transferring miR-21-5p and miR-155-5p [13], while miR-501-3p delivered by M2-Exos promoted the aggression of pancreatic ductal adenocarcinoma [14].

In this study, we demonstrated that M2-Exos carrying miR-21-5p augmented the migration and invasion of RCC cells by downregulating the phosphatase and tensin homolog (PTEN) tumor suppressor, which in turn activated the Akt pathway. The PTEN/Akt is a critical signaling axis involved in multiple cancer metastasis [15,16,17]. This study uncovered the critical contribution of exosomal miR-21-5p from M2 macrophages in promoting RCC progression and metastasis and a new potential target to dampen metastatic RCC.

## 2. Results

### 2.1. M2 Macrophages Promote RCC Migration and Invasion

M2 TAMs are a crucial component of the tumor microenvironment and promote tumor progression and metastasis [18,19]. We confirmed that RCC tumors contain M2 macrophages by performing immunohistochemistry (IHC) for CD163, a clinically-relevant marker of M2 macrophages [20], in ten consecutive surgical cases of RCC of different subtypes that were excised by one surgeon. CD163+ TAMs were present in all of them. Four representative IHC images are shown (Figure 1A).

To study the impact of macrophages on RCC tumor cells, we used the human THP-1 cell line. THP-1 is a widely-used human monocytic leukemia cell line that can be induced to a macrophage-like state (THP-1-Mφ) by phorbol ester (PMA) [21]. The THP-1-Mφ cells were further polarized to an M2 state (THP-1-M2) with IL4. Compared to THP-1-Mφ cells, THP-1-M2 cells expressed significantly higher markers of M2 polarization, CD163 and CD206, but unchanged expression of the M1-marker genes HLA-DR and TNF-α (Figure 1B). THP-1-M2 cellular morphology is elongated and spindle-shape, whereas THP-1-Mφ cells are round (Appendix A). Immunofluorescence (IF) staining revealed that the majority of THP-1-M2 cells express CD206 (Figure 1C). This data confirms that our polarization scheme successfully generates monocytes with the characteristics of M2 macrophages.

To investigate the pro-metastatic influence of M2 TAMs, we therefore evaluated whether soluble factors from M2 macrophages could augment aggressive behaviors of RCC tumor cells: migration and invasion. The 786-O (clear cell subtype) and ACHN cells (papillary subtype) were cultured in media conditioned by THP-1-Mφ cells (Mφ-CM) or THP-1-M2 cells (M2-CM), or control RPMI-1640 medium. Compared to control media or the Mφ-CM, the M2-CM significantly enhanced the vertical migration and invasion of RCC cells through a transwell assay (Figure 1D,F and Appendix A). Moreover, wound-healing assays also showed increased transverse migration of 786-O and ACHN cells in M2-CM compared to control media (Figure 1E,G). These findings suggest that M2 macrophages promote RCC migration and invasion via a soluble paracrine mechanism.

### 2.2. Characterization and Internalization of M2-Exos

Recent studies provide clear evidence that exosomes can facilitate cancer metastasis [12,13,14,22,23]. Therefore, we investigated whether M2-Exos could deliver pro-metastatic signals in RCC. We first analyzed the physical properties of M2-Exos. Transmission electron microscopy (TEM) revealed the morphology of M2-Exos to be vesicles about 100 nm in diameter (Figure 2A). Consistent with TEM findings, dynamic light scattering (DLS) also showed the diameters of M2-Exos to predominantly be between 60 and 140 nm (Figure 2B). Western blotting confirmed high levels of exosomal marker proteins, CD63 and CD81, in the M2-Exos preparation (Figure 2C).

Next, we investigated the cellular internalization of M2-Exos by labeling them with PKH67, a green, fluorescent membrane dye. The fluorescent-labeled exosome clusters were taken up by 786-O and ACHN cells (Figure 2D,F). The three-dimensional confocal construction further showed that these M2-Exos were located within the RCC cells (Figure 2E,G). The uptake of the exosome clusters increased over time, reaching a plateau at about 10 h (Figure 2H). Collectively, these results show that the physical characteristics of M2-Exos isolated from THP-1-M2 cells are consistent with prior studies and that they are effectively internalized by RCC tumor cells.

### 2.3. M2-Exos Promote RCC Migration and Invasion

To investigate whether M2-Exos could transmit the paracrine signaling that influences the metastatic behaviors, RCC cells were cultured with or without M2-Exos. Compared to the control group, the presence of M2-Exos markedly increased the invasion and migration of 786-O and ACHN cells in transwell and wound-healing assays, respectively (Figure 3A–D). Epithelial-mesenchymal transition (EMT), a developmental program, has been shown to confer metastatic properties to cancer cells by improving mobility and invasion [24,25]. To assess if EMT could contribute to the observed heightened invasive behavior, we examine the expression of EMT markers at the mRNA and protein levels using RT-qPCR and western blotting, respectively. The levels of two EMT markers, MMP-9 and vimentin, were significantly higher in the M2-Exos–treated group than the control group in both 786-O and ACHN cells (Figure 3E–H). Interestingly, M2-Exos did not promote proliferation in RCC cells (Figure 3I,J). Collectively, our finding indicated that M2-Exos could promote migration, invasion, and EMT of RCC cells.

### 2.4. MiR-21-5p in M2-Exos Directs RCC Aggressive Behaviors In Vitro and Metastasis In Vivo

Emerging evidence suggests that the impact of exosomes on cancer metastasis could be mediated by miRNAs [23,26,27]. Three exosomal miRNAs are most frequently involved in cancer migration and metastasis: miR-21-5p [13,28], miR-155-5p [13,29,30], and miR-210-3p [31,32]. Comparing the expression of these three miRNAs by RT-qPCR, miR-21-5p had the highest expression in M2-Exos (Figure 4A). To further verify that miR-21-5p is directing the pro-metastatic effects of M2-Exos, we employed an anti-miR inhibitor to specifically knock down miR-21-5p in THP-1-M2 cells by transfecting them with miR-21-5p inhibitor (miR-21-Inh) or miR-21-5p negative control (miR-21-NC). Exosomes from miR-21-Inh treated THP-1-M2 cells (miR-21-Inh-Exos) expressed a markedly lower level of miR-21-5p than the exosomes from miR-21-NC treated THP-1-M2 cells (miR-21-NC-Exos) (Figure 4B). The addition of miR-21-NC-Exos to both 786-O and ACHN cells resulted in a significantly higher intracellular level of miR-21-5p, 5.9- and 3.3-fold higher than control untreated cells, respectively (Figure 4C,D). On the other hand, the addition of miR-21-Inh-Exos resulted in significantly less increase than miR-21-NC-Exos (Figure 4C,D). The functional impacts of these treated exosomes paralleled their level of miR-21-5p assayed. Either by the protein expression level of EMT markers (MMP-9 and Vimentin) (Figure 4E,F), or vertical migration and invasion capability (Figure 4G,I), or transverse migration ability (Figure 4H,J), the miR-21-NC-Exos treated cells showed the greatest increase compared to the control untreated cells, while the miR-21-Inh-Exos group showed some increase, but this increase was always significantly less than the miR-21-NC-Exos group. Collectively, the data presented supports that miR-21-5p delivered by M2-Exos can enhance the migration and invasion capabilities of RCC cells in vitro. The depletion of miR-21-5p in M2-Exos by miR-21-Inh treatment was effective but incomplete, lowering the level of miR-21-5p in exosomes by 89.8% (Figure 4B). We surmise that the incomplete miR-21-5p depletion is responsible for the partial activity of miR-21-Inh-Exos.

Next, we sought to confirm this interaction in vivo. To do so, we established RCC xenografts in the chorioallantoic membrane (CAM) model [33,34]. Our experience has shown that the CAM model is an efficient, time-saving in vivo model, especially for establishing human RCC xenografts [35,36]. Furthermore, it faithfully recapitulates the metastatic behaviors of the mouse model [37,38]. Here, we utilized fertilized duck eggs rather than the traditional chicken eggs because they offered a more extended tumor growth/incubation period (up to a maximum of 18 days vs. 12 days) to assess metastatic potential [39]. The 13-day tumor growth period used in this study (Figure 5A) permits the detection of micrometastases in the embryonic organs by qPCR but is insufficient for detecting metastasis by histology. Three groups of 786-O cells were implanted into the CAM model on postfertilization day 12, after treatment with the exosomes-free complete media as a control (*n* = 5), miR-21-Inh-Exos (*n* = 5), or miR-21-NC-Exos (*n* = 6), respectively. Although the growth of 786-O cells caused mild hemorrhage of CAM, all three groups of tumors were established without difficulty and harvested on postfertilization day 25 (Appendix A). Detailed histological examination showed viable tumors in all three groups (Figure 5B). Treatment with different exosomes did not affect the tumor growth, as there were no significant differences in tumor weight (Figure 5C). The metastatic spread of 786-O cells to the embryonic liver was assessed by qPCR for the human albumin gene (ALB) normalized to the avian actin gene (ACTB). Compared to the control group, the level of metastatic spread was highest in the miR-21-NC-Exos treated group and significantly higher in the miR-21-Inh-Exos group (Figure 5D). The significant but incomplete depletion of miR-21-5p in miR-21-Inh-Exos diminished their metastasis-promoting activity compared to miR-21-NC-Exos, but they enhanced the metastatic potential over the control group. Taken together, these findings strongly support that the miR-21-5p delivered by M2-Exos can promote aggressive behaviors of RCC cells in vitro that lead to metastatic dissemination in vivo.

### 2.5. MiR-21-5p in M2-Exos Promotes RCC Metastasis through PTEN/Akt Signaling

A known mechanism of action for miRNAs is binding to the 3′-untranslated regions (3′UTRs) of genes to downregulate their expression, which occurs in the regulation of cancer metastasis [40]. PTEN is a well-known tumor suppressor gene that regulates Akt signaling to promote tumorigenesis and metastasis [15,16,17], including in RCC [41]. It has been reported to be a direct target of miR-21-5p in some malignancies [42,43]. Here, we assessed if PTEN could be the target of miR-21-5p driving RCC aggression. A potential miR-21-5p binding site was identified within the 3′-UTR of PTEN mRNA (PTEN-3′UTR; Figure 6A). We created a reporter construct linking the luciferase gene to the wildtype PTEN-3′UTR (PTEN-3′UTR-WT) or the PTEN-3′UTR containing a mutated target site (PTEN-3′UTR-Mut; Figure 6A). These reporters were cotransfected with the miR-21-Inh or miR-21-NC in 786-O and ACHN cells. When miR-21-5p was downregulated, the PTEN-3′UTR-WT reporter showed increased luciferase activity, but the PTEN-3′UTR-Mut construct did not (Figure 6B,C). These results support that miR-21-5p targets the 3′UTR of PTEN mRNA in RCC cells.

We next directly analyzed the impact of miR-21-Inh-Exos or miR-21-NC-Exos on PTEN and AKT expression. The addition of miR-21-NC-Exos to 786-O and ACHN cells resulted in significant downregulation of PTEN mRNA compared to treatment with miR-21-Inh-Exos (Figure 6D,F). This lowered PTEN expression corresponded to increased activation of Akt as measured by phosphorylation at serine 473 (Figure 6E,G). To confirm that PTEN suppression could promote metastatic behaviors of miR-21-Inh-Exos, we inhibited PTEN with a phosphatase inhibitor (SF1670). PTEN inhibition significantly promoted migration and invasion in 786-O or ACHN cells treated with miR-21-Inh-Exos (Figure 6H–K). Taken together, our results indicate that the miR-21-5p delivered by M2-Exos regulates the PTEN/Akt oncogenic pathway to promote RCC metastasis.

## 3. Discussion

In this study, we investigated how M2 macrophages could enhance the metastatic behaviors of RCC tumor cells. We discovered that the M2-Exos harvested from conditioned media of M2 macrophages is the culprit that boosted the migration and invasion capabilities of RCC cells. Moreover, miR-21-5p was identified as the payload in M2-Exos that promoted the pro-metastatic effects in vitro and distant organ metastasis in vivo in the duck CAM xenograft model. The miR-21-5p targeted the PTEN-3′UTR to downregulate this tumor suppressor and activate Akt signaling to enhance tumor cells migration and invasion. The proposed M2-Exos-mediated pro-metastatic mechanism is summarized in Figure 7. Further, we showed that using a specific miR-21-5p anti-miR inhibitor could counter the pro-metastatic effects. This work opens a new therapeutic window to lower the pro-tumorigenic influence of M2 TAMs.

M2 macrophages have been extensively studied for the multitude of different mechanisms they contribute to cancer metastasis [8,9]. For instance, Lee et al. reported that TAMs, prominently expressing M2 markers, could secrete interleukin-35 to enhance metastatic colonization of cancer cells [44]. CHI3L1 secreted by M2 TAMs might activate the mitogen-activated protein kinase signaling pathway to promote the metastasis of breast and gastric cancer cells [19]. Here, we demonstrated that a different soluble mediator produced by the M2 macrophages, namely exosomes, could promote the metastatic potential of RCC cells.

A large volume of recent studies emphasizes the importance of exosomes in cancer progression [12,13,14,22,23,45]. In particular, M2-Exos have been well-recognized as essential mediators of intercellular communication to promote cancer metastasis in different types of cancer. For instance, Apolipoprotein E protein in M2-Exos could encourage the dissemination of gastric cancer cells [46]. In esophageal cancer, the metastatic potential of tumor cells was upregulated by M2-Exos via lncRNA AFAP1-AS1 [47]. In another study, M2-Exos were reported to enhance the invasion and metastasis of hepatocellular carcinoma cells by transferring α_M_ β_2_ integrin [48]. The work presented in this study demonstrates an additional mechanism M2-Exos employ to promote RCC aggression via miRNA.

Interestingly, miR-21-5p delivered by exosomes has been identified to play an instrumental role in cancer metastasis. For example, exosomal miR-21-5p derived from gastric cancer cells could induce EMT to promote peritoneal metastasis [28]. Our discovery of miR-21-5p as the pro-metastatic exosome-mediated directive from TAMs to RCC tumor cells represents an exciting finding to build further investigations. The importance of PTEN/Akt signaling in cancer progression and metastasis is fully substantiated in many malignancies [15,17]. PTEN has been reported as the target of miR-21-5p [42,49], but this is not a widely recognized finding. We identified the specific sequence in PTEN-3′UTR that miR-21-5p is targeting and clearly showed the impact of downregulating PTEN expression and activating Akt signaling to promote distant metastasis in RCC. In total, we showed that M2-Exos with miR-21-5p delivered to RCC tumor cells promotes pro-metastatic behavior in vitro and distant metastatic dissemination in vivo. Importantly, depleting miR-21-5p in macrophages by the anti-miR reversed this pro-metastatic drive. This work establishes a new avenue to investigate a well-known heterotopic metastatic crosstalk and offers a new therapeutic option to prevent or treat metastatic RCC.

There are several limitations to our study. First, the TEM image showed the M2-Exos as saucer-shaped vesicles. However, this morphology could be an artifact of the sample preparation techniques. Cryo-TEM results have shown that exosomes have a perfectly spherical structure in an aqueous solution [50]. Second, the M2-Exos-mediated pro-metastatic mechanism in this study is based on an in vitro M2 macrophage model system. Further validation of our findings in other systems, such as primary cells or patient-derived tumor tissues, will be very valuable. Third, we believe that there are other nucleic acid or protein payloads in M2-Exos besides miR-21-5p that could contribute to the metastatic behaviors we observed. Therefore, further studies are clearly warranted to enrich our understanding of the crosstalk mechanisms mediated by M2-Exos to promote RCC progression and metastasis.

## 4. Materials and Methods

### 4.1. Cell Culture

Human cell lines 786-O (ATCC, #CRL-1932) and ACHN (ATCC, #CRL-1611) were cultured in RPMI 1640 medium (Gibco, Waltham, MA, USA, #11875093) supplemented with 10% [*v*/*v*] heat-inactivated FBS (GenClone, #25-550), 2 mM L-glutamine (Corning, Corning, NY, USA, #25-005-CI), penicillin (100 IU/mL) and streptomycin (100 μg/mL) (Gibco, #15140-122). Human monocyte cell line THP-1 (ATCC, #TIB-202) was a gift from Professor Genhong Cheng’s lab (UCLA, Los Angeles, CA, USA). THP-1 cells were cultured in RPMI 1640 supplemented with 10 mM hepes (Gibco, #15630080), 1 mM pyruvate (Gibco, #11360070), 2.5 g/l D-glucose (Sigma-Aldrich, #G7021) and 50 pM β-mercaptoethanol (Sigma-Aldrich, Waltham, MA, USA, #M6250). To obtain Mφ subtype of THP-1 (THP-1-Mφ), THP-1 cells treated with 150 nM phorbol 12-myristate 13-acetate (PMA, Fisher BioReagents, Waltham, MA, USA, #BP685-1) for 48 h. THP-1-Mφ cells were then polarized into M2 subtype (THP-1-M2) with 48 h incubation of 20 ng/mL interleukin 4 (IL-4, Biolegend, San Diego, CA, USA, #574004).

### 4.2. Exosomes Isolation and TEM

To isolate exosomes from culture supernatant, fresh RPMI 1640 containing 10% exosomes-depleted FBS (bovine exosomes were removed by overnight centrifugation at 150,000× *g*, 16 h) was added to cells at 80% confluence. After 48 h incubation, the culture supernatant was sequentially centrifuged at 300× *g* for 5 min, 2000× *g* for 10 min, and 10,000× *g* for 30 min to eliminate cells and debris. Further, the supernatant was ultracentrifuged at 100,000× *g* for 70 min in an SW32Ti swing bucket rotor (Beckman Coulter L-100XP, Brea, CA, USA) at 4 °C. Exosomes pellets were washed by cold PBS and collected by ultracentrifugation again at 100,000× *g* for 70 min at 4 °C. Finally, exosomes were dissolving with PBS and stored at −80 °C. The size of exosomes was measured using Dynamic Light Scattering (DLS) as previously described by Lyu [51].

The exosomes morphology was identified by an FEI T20 transmission electron microscope (FEI Company, Hillsboro, OR, USA). A 5 µL drop of exosomes suspension (300 μg/mL) was applied onto a glow-discharged grid coated with carbon film (Ted Pella, #01814-F) and left on the grid for 30 s, followed by negative staining with 2% uranyl acetate. The TEM images of isolated exosomes were identified at 29,000× magnification in an FEI Tecnai TF20 electron microscope operated at 200 kV.

### 4.3. Internalization of PKH67-Labeled M2-Exos

Purified M2-Exos were suspended in a mixture of 500 µL of diluent C and 2 µL of PKH67 (MilliporeSigma, #MINI67) for 5 min at room temperature. The mixture was then incubated with an equal volume (500 µL) of 1% bovine serum albumin for 1 min followed by ultracentrifugation at 100,000× *g* for 70 min at 4 °C. The control group had only PKH67 mixed in diluent C without exosomes, and the other procedures were the same. The pellets were suspended with PBS and then used immediately or stored at −80 °C. Cells were treated with PKH67-labeled M2-Exos for 24 h at 37 °C and fixed with 4% paraformaldehyde. Cell nuclei were stained with 4′,6-diamidino-2-phenylindole (DAPI, Invitrogen, #D1306), and F-actin was monitored by rhodamine-phalloidin (Invitrogen, #R37112) staining. Fluorescence images were obtained on a fluorescence microscope (Nikon ECLIPSE Ti, Tokyo, Japan), while confocal images were acquired and converted into three-dimensional images with a confocal microscope (Carl Zeiss, LSM880, Oberkochen, Germany)”.

### 4.4. RNA Isolation and RT-qPCR

Total RNA was extracted using Trizol LS (Invitrogen, #10296010, Carlsbad, CA, USA) and was reverse-transcribed into cDNA using the PrimeScript RT Reagent Kit (TaKaRa, #RR037A, Kusatsu, Japan). The cDNA was amplified using the SensiMix SYBR & Fluorescein Kit (Bioline, #QT615-05) on the Quantstudio5 instrument (Applied Biosystems, Foster City, CA, USA). Specific primers (Appendix A) were purchased from Integrated DNA Technologies. The relative expression levels of mRNA or miRNA were normalized to glyceraldehyde-3-phosphate dehydrogenase (GAPDH) or U6 snRNA expression, respectively, using the 2−ΔΔCt method.

### 4.5. Western Blotting

Cells or exosomes were lysed in the RIPA buffer. Protein concentration was measured using the bicinchoninic acid (BCA) protein assay kit (Thermo Fisher Scientific, #A53226, Waltham, MA, USA). The loading protein was separated by SDS/PAGE gels and transferred to Immobilon-P (Millipore) membranes. After being blocked by 5% (w/v) BSA at room temperature for 1 h, the membranes were incubated with relevant primary antibodies and peroxidase-conjugated secondary antibodies (Jackson ImmunoResearch). The images were acquired by enhanced chemiluminescence (Millipore, #P90719, Billerica, MA, USA) with a Bio-Rad ChemiDoc XRS+ imaging system. The primary antibodies are as follows: β-actin (Santa Cruz Biotechnology, #sc-47778, 1:2000), MMP-9 (Abcam, #ab38898, 1:1000), vimentin (Abcam, #ab73843, 1:1000), PTEN (Cell Signaling Technology, #9559, 1:1000), Akt (pan) (C67E7) (Cell Signaling Technology, #4691, 1:1000), Phospho-Akt (Ser473) (Cell Signaling Technology, #4075, 1:1000), CD81 (Novus Biologicals, #NB100-65805, 1:1000), and CD63 (Novus Biologicals, #NB100-77913, 1:1000).

### 4.6. Transwell and Wound-Healing Assays

Transwell assays were performed in 24-well membrane inserts (Corning, #3422) with or without precoated Matrigel (Corning, #354234). 3 × 10^5^ of cells were seeded into upper chambers. After 24 h, noninvasive cells were removed by wiping with cotton swabs. Invading cells at lower surfaces of chamber membranes were fixed with methanol and stained with 1% crystal violet. Four random fields were counted under a light microscope, and each experiment was repeated three times.

For the wound-healing assay, cells with the full confluence in twelve-well plates were scratched with 200 uL pipette tips to draw gaps. Cells were cultured in a complete medium without FBS or additional conditions specified and observed by microscopy at specific time points. The migration distance was determined by the difference between wound widths.

### 4.7. Cell Proliferation Assay

Cell proliferation was measured by the MTS assay (Promega, #G3581) following the manufacturer’s protocol. Cells in each well of 96-well plates were incubated with 200 µL of MTS solution for 1 h at 37 °C. Optical density at the 490 nm wavelength was measured by a multi-detection spectrophotometer (CLARIOstar, BMG LABTECH, Offenburg, Germany).

### 4.8. Cell Transfection and miRNA Targeted Luciferase Reporter Assay

M2 macrophages were transfected with miR-21-5p inhibitor (Sigma-Aldrich, #HSTUD0371) (miR-21-Inh) or miR-21-5p negative control (Sigma-Aldrich, #HMC0002) (miR-21-NC) by lipofectamine RNAiMAX transfection reagent (Invitrogen, #13778100). Tumor cells were transfected with modified psiCHECK2 luciferase reporter plasmids by FuGENE HD transfection reagent (Promega, #E2311).

A binding sequence of miR-21-5p in PTEN-3′UTR (wild type) and its related mutant form (mutant type) were synthesized as DNA oligonucleotides (PTEN-3′UTR-WT and PTEN-3′UTR-Mut, respectively) (Appendix A). The two DNA oligonucleotides were subcloned into the dual-luciferase reporter vector (psiCHECK2, Promega) by the NotI and XhoI restriction enzymes (psiCHECK2-PTEN-3′UTR-WT and psiCHECK2-PTEN-3′UTR-Mut, respectively). These two luciferase reporters were co-transfected with miR-21-Inh or miR-21-NC into 786-O and ACHN cells, and luciferase activity was measured by the dual-luciferase assay (Promega, #E1910) after 48 h.

### 4.9. IHC and IF

For IHC, clinical RCC tissues were embedded in paraffin after fixation with 10% formalin and cut into sections of 4 mm thickness. Sections were deparaffinized, rehydrated, retrieved for antigen, and then incubated with 3% H2O2 for 10 min to inactivate endogenous peroxidase. After nonspecific bindings were blocked by 5% goat serum (Sigma-Aldrich, #G9023), the tissues were incubated with CD163 antibodies (Biocare Medical, #CM353AK, 1:400) and then HRP-conjugated secondary antibodies followed by the DAB Chromogen kit (Biocare Medical, #DB801L) staining. Finally, the tissues were stained with hematoxylin. The IHC images were obtained with a Nikon H600L microscope.

For IF, macrophages were fixed with 10% formalin and permeabilized with 0.1% Triton X-100 (Sigma-Aldrich, #9036-19-5). After being blocked by 5% goat serum at room temperature, macrophages were incubated with FITC-conjugated anti-human CD206 antibody (BioLegend, #321104, 1:200, San Diego, CA, USA) at 4 °C overnight. Cell nucleus was stained with DAPI, and cytoskeletal F-actin was labeled with rhodamine-phalloidin. Fluorescence images were visualized by a fluorescence microscope (Nikon ECLIPSE Ti, Tokyo, Japan).

### 4.10. CAM Model and Quantitative PCR (qPCR) Analysis of Genomic DNA

To assess RCC xenografts metastasis in vivo, we established the CAM model with fertilized duck eggs as our previously published protocols [35,36,37,38]. Freshly laid fertilized duck eggs were incubated in a MultiQuip incubator (MULTIQUIP, E2, AISTRAL, Australia) at 37 °C with 60% humidity. On postfertilization day 10, a window with a diameter of 2 cm was drilled into the eggshell to lower the CAM and create an air pocket. On postfertilization day 12, 3 × 10^6^ 786-O cells per egg were diluted in 50 µL of RPMI-1640 and 50 µL of Matrigel and implanted on the CAM. On postfertilization day 25, the embryos were euthanized using isoflurane. The CAM tumors were collected for tumor weight assessment.

To detect the metastasis of human RCC cells in CAM models, genomic DNA was extracted from duck embryo livers using the standard phenol extraction method. Then qPCR analysis for human ALB (TaqMan Gene Expression Assay, Applied Biosystems, Hs99999922_s1, #4331182) normalized to avian ACTB (TaqMan Gene Expression Assay, Applied Biosystems, Gg03815934_s1, #4331182) was performed. The qPCR reaction mixture of a 20 µL final volume contained 0.5 µL TaqMan Gene Expression Assay, 1 uL DNA sample, 3.5 µL nuclease-free water, and 5 µL TaqMan fast advanced master mix (2×) (Applied Biosystems, #4444557). The qPCR was performed under the following conditions: UNG enzyme incubation at 50 °C for 120 s and polymerase activation at 95 °C for 20 s followed by 40 cycles of 1 s at 95 °C and 20 s at 60 °C.

### 4.11. Statistics Analysis

GraphPad Prism version 9.0.0 was used to perform statistical analysis. The quantitative data were shown as mean ± standard deviation. Two-tailed tests were used to analyze differences between groups. Asterisks indicate significant statistical differences (* *p* < 0.05, ** *p* < 0.01, *** *p* < 0.001, **** *p* < 0.0001).

## 5. Conclusions

In conclusion, our study revealed that M2-Exos delivers miR-21-5p to RCC tumor cells to promote their metastatic potential via PTEN/Akt signaling. Hence, downregulation of miR-21-5p could be a novel approach to reduce the pro-metastatic influences of M2 macrophages and offers a better prognosis for patients with metastatic RCC.

## Figures and Tables

**Figure 1 ijms-23-03005-f001:**
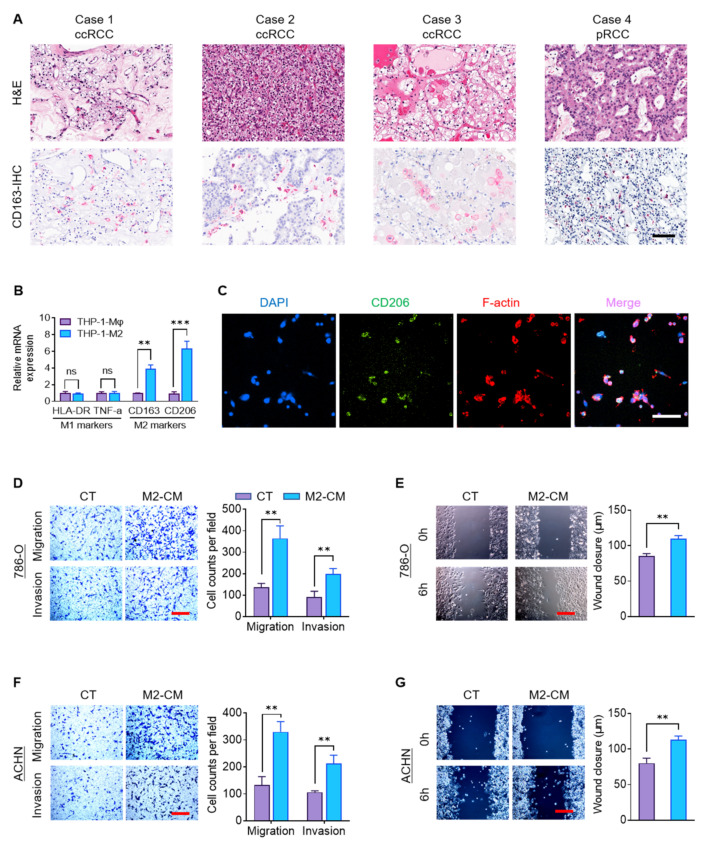
M2 macrophages promote RCC migration and invasion. (**A**) H&E and CD163 IHC staining (red) of four representative clinical RCC samples. Scale bar: 50 μm. (**B**) Macrophage marker mRNA expression was assayed in THP-1-Mφ and THP-1-M2 cells by RT-qPCR. (**C**) IF staining of nuclei (DAPI), CD206, and F-actin in THP-1-M2 cells. Scale bar: 50 μm. (**D**–**G**) Cellular migration and invasion of 786-O and ACHN cells treated with M2-CM or control media were assessed by transwell and wound-healing assays. Scale bar: 100 μm. ** *p* < 0.01, *** *p* < 0.001, ns: not significant. pRCC: papillary renal cell carcinoma; CT: control; RT-qPCR: reverse transcription-quantitative PCR.

**Figure 2 ijms-23-03005-f002:**
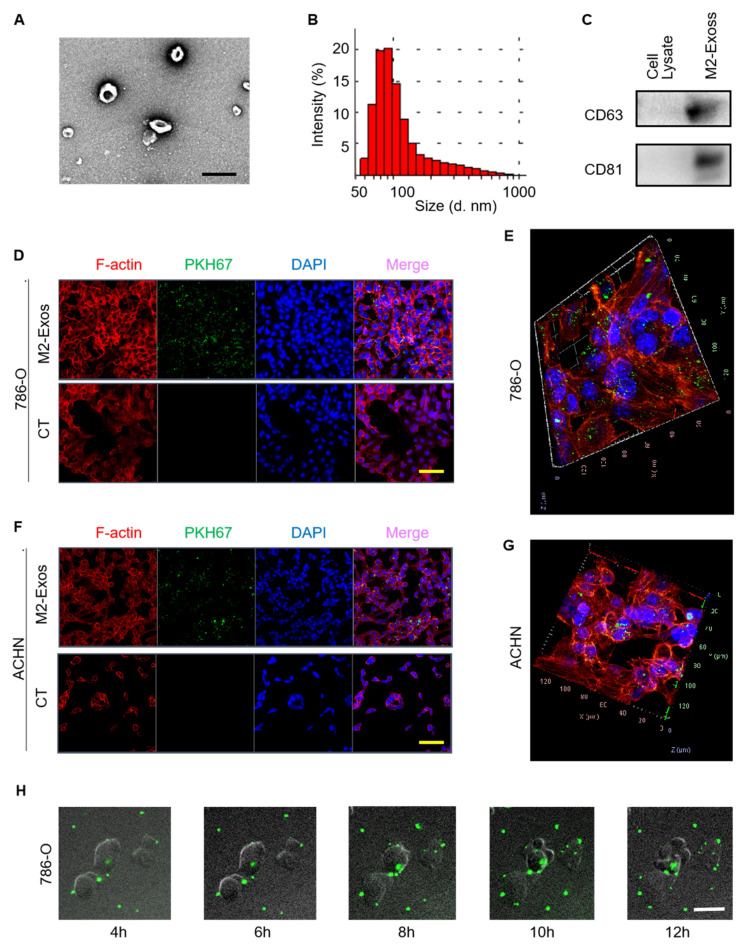
Characterization and internalization of M2-Exos. (**A**) TEM image of M2-Exos. Scale bar: 200 nm. (**B**) DLS measurement of M2-Exos size. (**C**) Western blotting assay of exosomal markers in THP-1-M2 cellular lysate and M2-Exos preparation. (**D**,**F**) Fluorescence images of 786-O and ACHN cells treated with or without PKH67-labeled M2-Exos (green). Scale bar: 50 μm. (**E**,**G**) Three-dimensional confocal reconstruction of 786-O and ACHN cells treated with PKH67-labeled M2-Exos (green). (**H**) Fluorescence staining analyzing the internalization of M2-Exos by 786-O cells over 12 h. Scale bar: 10 μm. CT: control.

**Figure 3 ijms-23-03005-f003:**
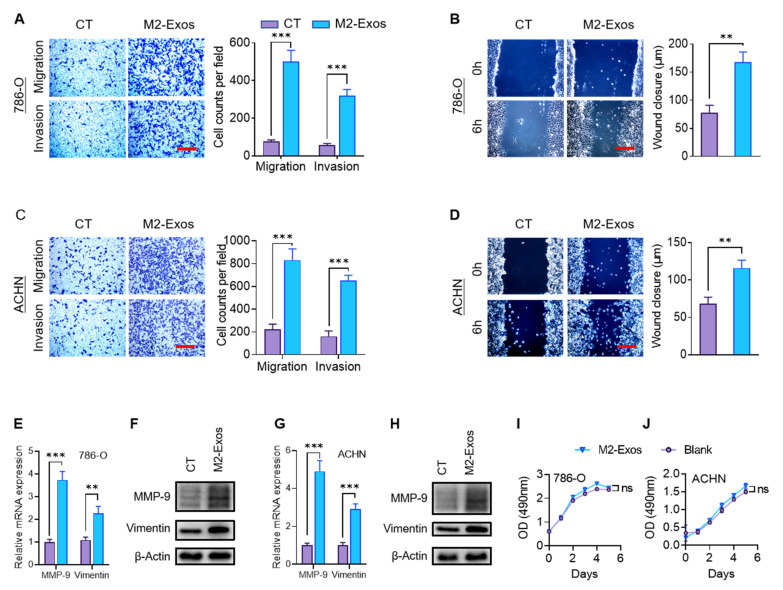
M2-Exos promote RCC migration and invasion. 786-O and ACHN cells treated with or without M2-Exos. (**A**–**D**) The impact of the M2-Exos treatment on the migration and invasion capabilities of RCC cells was analyzed by transwell (**A**,**C**) and wound-healing assays (**B**,**D**). Scale bar: 100 μm. (**E**,**F**) The impact of M2-Exos treatment on RCC cell: mRNA (**E**,**G**) and protein expression (**F**,**H**) of MMP-9 and vimentin were assayed by RT-qPCR and western blotting, respectively. (**I**,**J**) The impact of M2-Exos treatment on cell proliferation was analyzed by MTT assay. ** *p* < 0.01, *** *p* < 0.001. CT: control.

**Figure 4 ijms-23-03005-f004:**
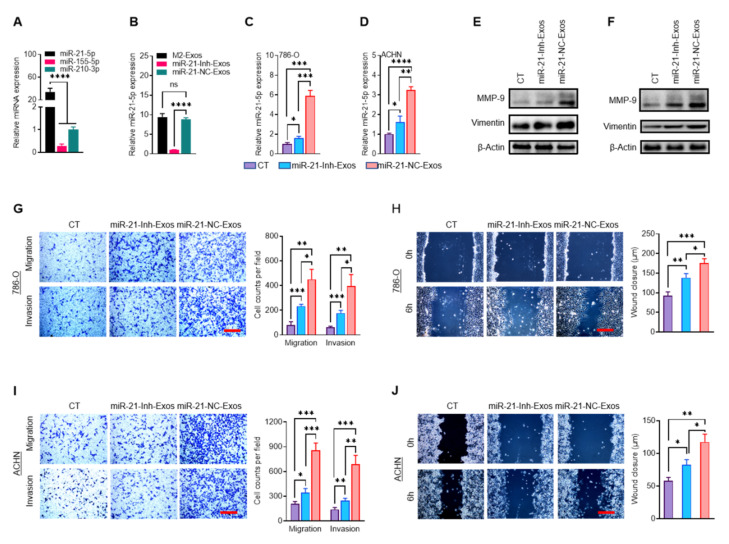
MiR-21-5p in M2-Exos promotes RCC migration and invasion in vitro. (**A**) The expression of three miRNAs in THP-1-M2 cells was analyzed by RT-qPCR. (**B**) The level of miR-21-5p expression in M2-Exos, M2-miR-21-Inh-Exos, and M2-miR-21-NC-Exos was assessed by RT-qPCR. (**C**,**D**) The intracellular content of miR-21-5p in 786-O and ACHN cells treated with control media, miR-21-Inh-Exos, or miR-21-NC-Exos was assessed by RT-qPCR. (**E**,**F**) The level of MMP-9 and vimentin in treated cells was assessed by western blotting. (**G**–**J**) The migration and invasion of treated cells were assessed by transwell (**G**,**I**) and wound-healing assays (**H**,**J**). Scale bar: 100 μm. * *p* < 0.05, ** *p* < 0.01, *** *p* < 0.001, **** *p* < 0.0001, ns: not significant.

**Figure 5 ijms-23-03005-f005:**
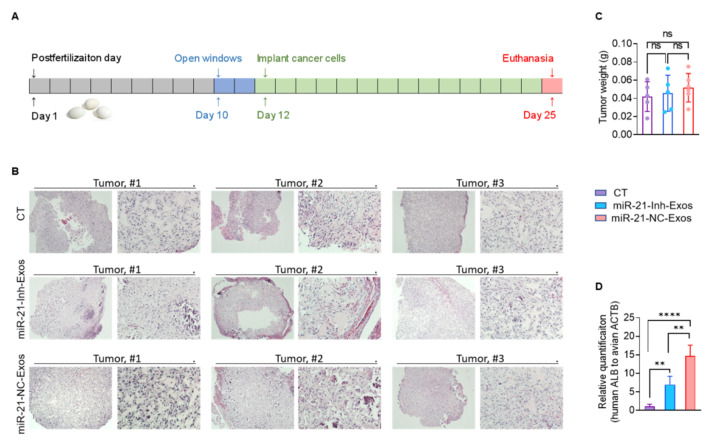
MiR-21-5p in M2-Exos directs RCC aggressive behaviors in vivo. (**A**) Schematic diagram of the timeline of CAM xenograft implantation. Three groups of 786-O tumor cells were implanted on CAMs following treatment with exosomes-free media (the control group), miR-21-Inh-Exos, or miR-21-NC-Exos. (**B**) H&E stains of three representative CAM-derived tumors from each group are shown. Magnification: 40×, 200×, for each tumor sample. (**C**) The tumor weights of harvested CAM tumors from each group. (**D**) qPCR for human ALB in embryonic livers normalized to avian ACTB. ** *p* < 0.01, **** *p* < 0.0001, ns: not significant.

**Figure 6 ijms-23-03005-f006:**
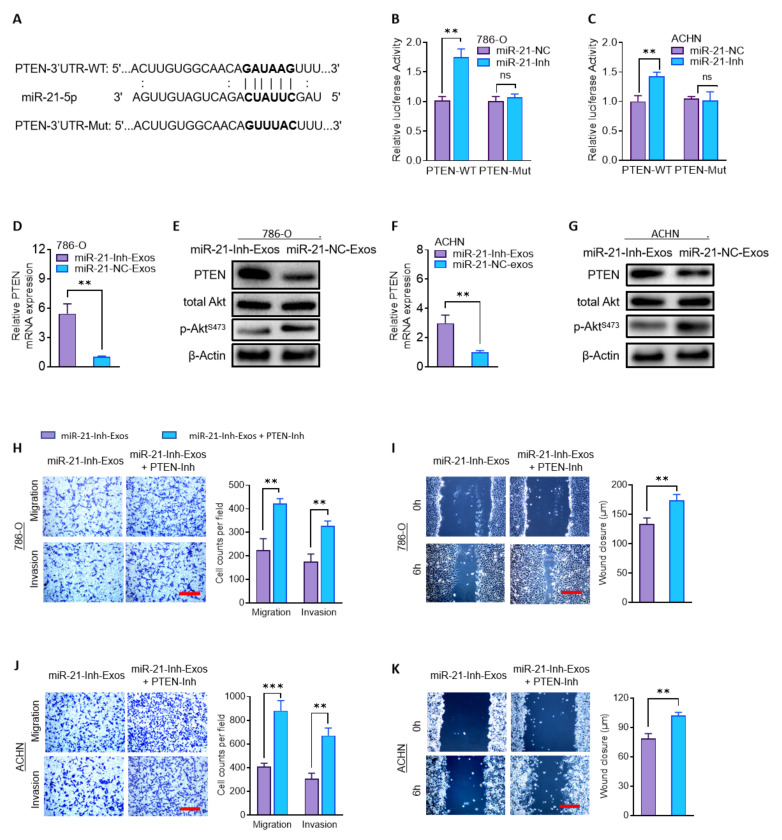
MiR-21-5p in M2-Exos regulates PTEN/AKT signaling to promote RCC metastasis. (**A**) The sequences of the putative target site for miR-21-5p in the PTEN-3′UTR (PTEN-3′UTR-WT) and a mutated version of this target (PTEN-3′UTR-Mut) were shown. (**B**,**C**) 786-O and ACHN cells were co-transfected psiCHECK2-PTEN-3′UTR-WT or psiCHECK2-PTEN-3′UTR-Mut, with miR-21-Inh or miR-21-NC. The relative luciferase activities in the cells were assayed. (**D**–**G**) The expression of PTEN mRNA, and protein level of PTEN, total Akt, and p-AktS473 were assessed in 786-O and ACHN cells treated with miR-21-Inh-Exos or miR-21-NC-Exos. (**H**–**K**) The migration and invasive capabilities of cells treated with miR-21-Inh-Exos or miR-21-Inh-Exos + PTEN inhibitor (PTEN-Inh) were analyzed by transwell and wound-healing assays. Scale bar, 100 μm. ** *p* < 0.01, *** *p* < 0.001, ns: not significant.

**Figure 7 ijms-23-03005-f007:**
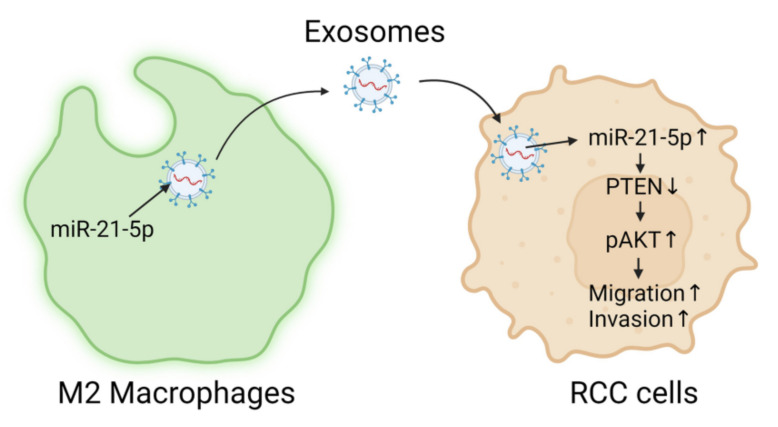
Schematic diagram of the proposed mechanism of M2-Exos mediated enhancement of RCC aggression via miR-21-5p.

## Data Availability

The data used or analyzed during the current study are available from the corresponding author.

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
