# Peer review of "The miRNA-21-5p Payload in Exosomes from M2 Macrophages Drives Tumor Cell Aggression via PTEN/Akt Signaling in Renal Cell Carcinoma"

_ijms, 2022, doi:10.3390/ijms23063005_

Round 1

Reviewer 1 Report

Very well documented and strightly performed cancer research paper showes that macrophage 2-subtype of exos (M2-exos) are involved in migration and invasion capabilities of renal cancer cells.  Both in in vitro and in vivo systems Authors demonstrate the mechanism of finally cancer dissemination is depending on M2-exos carrying miR-21-5p which downregulates PTEN-tumor suppressor gene to promote Akt signalling  tumorigenesis and metastasis.

Figure 7 sumarizes the mechanism of well-done research.

some punctuation errors in references

Reviewer 2 Report

Major comments:

I believe the experiment shown in Figure 2H do not fully support the conclusions of the authors. The exosome clusters could very well just be moving around in the media and have been captured in that particular spot. Higher-resolution confocal imaging would be necessary to make such claims, and/or a thorough media wash to verify that exosomes remain.

Minor comments:

The shape of exosomes in TEM could be attributed to the sample preparation techniques. I believe this conclusion is not fully supported by data. I believe this statement should be requalified.

Please, double-check the scalebar in figure 2 D-H. 786 cells are in the scale of several microns in size, and therefore should be much bigger than a scalebar of 200 nm. Further, I believe authors should be careful to qualify exosomes as visible at the presented magnification. Further, this reviewer believes the observed particles are clusters of exosomes that grouped during the staining procedure. I believe the authors should clarify that section as well.

Please, qualify that the mechanism presented in figure 7 is a proposal since it should be validated in other model systems (e.g., primary cells, other cell lines), and may not be the only mechanism responsible for the observed behaviors.

Please, include a short paragraph indicating current limitations or your study.
